# Novel anti-malarial drug strategies to prevent artemisinin partner drug resistance: A model-based analysis

**Amber Kunkel** [1]*, **Michael White** [2], **Patrice Piola** [3]

**1** Emerging Diseases Epidemiology Unit, Institut Pasteur, Paris, France, **2** Malaria: Parasites and Hosts Unit, Institut Pasteur, Paris, France, **3** Epidemiology Unit, Institut Pasteur du Cambodge, Phnom Penh, Cambodia

* agkunkel@gmail.com

**Data Availability Statement:** Codes used to create manuscript are available at https://github.com/agkunkel/cambodia-partner-drugs.

**Funding:** AK was supported by the Pasteur Foundation (US). The funders had no role in study

## Abstract

Emergence of resistance to artemisinin and partner drugs in the Greater Mekong Subregion has made elimination of malaria from this region a global priority; it also complicates its achievement. Novel drug strategies such as triple artemisinin combination therapies (ACTs) and chemoprophylaxis have been proposed to help limit resistance and accelerate elimination. The objective of this study was to better understand the potential impacts of triple ACTs and chemoprophylaxis, using a mathematical model parameterized using data from Cambodia. We used a simple compartmental model to predict trends in malaria incidence and resistance in Cambodia from 2020–2025 assuming no changes in transmission since 2018. We assessed three scenarios: a status quo scenario with artesunate-mefloquine (ASMQ) as treatment; a triple ACT scenario with dihydroartemisinin-piperaquine (DP) plus mefloquine (MQ) as treatment; and a chemoprophylaxis scenario with ASMQ as treatment plus DP as chemoprophylaxis. We predicted MQ resistance to increase under the status quo scenario. Triple ACT treatment reversed the spread of MQ resistance, but had no impact on overall malaria incidence. Joint MQ-PPQ resistance declined under the status quo scenario for the baseline parameter set and most sensitivity analyses. Compared to the status quo, triple ACT treatment limited spread of MQ resistance but also slowed declines in PPQ resistance in some sensitivity analyses. The chemoprophylaxis scenario decreased malaria incidence, but increased the spread of strains resistant to both MQ and PPQ; both effects began to reverse after the intervention was removed. We conclude that triple ACTs may limit spread of MQ resistance in the Cambodia, but would have limited impact on malaria incidence and might slow declines in PPQ resistance. Chemoprophylaxis could have greater impact on incidence but also carries higher risks of resistance. Aggressive strategies to limit transmission the GMS are needed to achieve elimination goals, but any intervention should be accompanied by monitoring for drug resistance.

design, data collection and analysis, decision to
publish, or preparation of the manuscript.

**Competing interests:** The authors have declared
that no competing interests exist.

## Author summary

Artemisinin combination therapies (ACTs) consisting of an artemisinin derivative plus a partner drug are used to treat malaria worldwide. In Cambodia, resistance to artemisinin is widespread, and resistance to the partner drugs mefloquine and piperaquine has also emerged. We used a mathematical model to compare two strategies with the current status quo in Cambodia: first, a triple ACT scenario in which first-line treatment is an artemisinin derivative combined with two different partner drugs, and second, a chemoprophylaxis scenario in which one ACT is used for first-line treatment and a separate one is used as chemoprophylaxis. The triple ACT scenario limited the spread of mefloquine resistance but had minimal impact on the number of malaria cases. In some sensitivity analyses, it also slowed declines in piperaquine resistance. Chemoprophylaxis reduced the number of malaria cases and increased resistance, but both of those effects were short-lived. We conclude that triple ACTs may prevent the spread of partner drug resistance, but could be less effective against pre-existing resistance in the population. Additionally, triple ACTs would need to be coupled with other interventions to decrease cases. Chemoprophylaxis could immediately reduce malaria transmission, but risks include spread of resistance and a post-intervention rebound in cases.

## Introduction

Previous progress towards malaria elimination was lost and millions died when resistance to chloroquine and sulfadoxine-pyrimethamine emerged in the Greater Mekong Subregion (GMS) and spread to Africa [1]. Artemisinin Combination Therapies (ACTs) consisting of an artemisinin derivative plus a partner drug with longer half-life (ex. mefloquine, piperaquine, lumefantrine) were initially thought to be less prone to resistance due to rapid parasite clearance and multiple mechanisms of action [2]. However, slow clearance of malaria parasites caused by artemisinin resistance and failure of ACTs caused by subsequent partner drug resistance has now been reported in the GMS [3]. A study conducted from 2015–2018 in Thailand, Vietnam, and Cambodia found the efficacy of dihydroartemisinin-piperaquine (DP) at day 42 was only 50%, and over 90% of patients' samples showed mutations in the kelch13 gene associated with artemisinin resistance [4]. If artemisinin resistance were also to spread to Africa, it could lead to drastic increases in malaria mortality owing to the use of parenteral artesunate in severe malaria [5]. Though a spread in artemisinin partner drug resistance could be easier to manage due to the existence of multiple alternatives, modeling work suggests it could result in greater increases in transmission and incidence of clinical malaria than spread of artemisinin resistance alone [6].

National malaria control programs in the GMS thus face a challenging paradox. On the one hand, the presence of resistance to artemisinin and partner drugs makes malaria elimination in this region absolutely paramount [3,7]. At the same time, it also makes malaria control and elimination more challenging by reducing the efficacy of first-line treatment. In Cambodia, resistance has prompted two changes in first-line treatment since the introduction of ACTs in the early 2000s, from artesunate-mefloquine (ASMQ) to DP (around 2008–2010) and back again (around 2017), and there are fears that the current efficacy of ASMQ could be short-lived [8].

The challenge of malaria elimination in the presence of resistance to artemisinin and partner drugs has prompted the consideration of novel drug strategies in the GMS. Two strategies that have received particular attention are the use of triple ACTs containing two partner drugs,

for example, dihydroartemisinin-piperaquine plus mefloquine and artemether-lumefantrine plus amodiaquine [9] and chemoprophylaxis for high-risk individuals such as forest goers [10,11]. Previous models have shown that applying novel drug strategies such as multiple first line treatments could prevent emergence and spread of anti-malarial drug resistance [12,13]. Similarly, models of other diseases such as have shown the potential implications of combination therapies and chemoprophylaxis on drug resistance [14,15]. Although models have investigated the within-host implications of triple ACTs, however, the potential impact of these strategies on antimalarial drug resistance in the population has not yet been assessed [16].

The purpose of this paper was to use a simple model of malaria transmission to assess the mechanisms through which triple ACTs and chemoprophylaxis could affect artemisinin partner drug resistance in the GMS, using Cambodia as a motivating example. We focus on partner drug resistance, rather than resistance to artemisinin, as artemisinin resistance is already widespread in Cambodia and, on its own, rarely leads to treatment failure [17,18].

## Results

We created a simple compartmental model of malaria transmission and resistance to mefloquine (MQ) and piperaquine (PPQ). We fit the model separately to data from Eastern and Western Cambodia, as these two regions have seen different resistance trends and containment policies. Uncertain parameters were inferred based on trends in malaria cases, prevalence, and resistance from 2000–2018. However, we assumed that all parameters except those related to the interventions of interest remained fixed from 2018 onwards; this does not accurately reflect the impressive recent progress towards malaria elimination in Cambodia since 2018, but allows us to more clearly understand the effects of the different drug policies when other interventions are held constant.

We evaluated different strategies for use of MQ and PPQ beginning in 2020. Under the first, "status quo" scenario, ASMQ is used as first line treatment and DP is not used. Second, we evaluated a "triple ACT" scenario in which dihydroartemisinin-piperaquine plus mefloquine (DP-MQ) is used as first line treatment. Finally, we compared the results of these two scenarios with a third "chemoprophylaxis" scenario, in which ASMQ is used as first line treatment and DP is applied as chemoprophylaxis. We sought to determine the impact of these different strategies on 1) spread of partner drug resistance (with a particular emphasis on resistance to both drugs), and 2) progress towards malaria elimination.

### Baseline results

The model was able to capture general trends in malaria reported cases, prevalence, and resistance to artemisinin partner drugs from Cambodia from 2000–2019. These plots are shown in Section 2 of the S1 Appendix.

Fig 1 shows the results of the model under the baseline assumptions and best-fitting parameter sets, with interventions added in 2020. Compared to the status quo scenario, the triple ACT scenario produced very similar overall malaria incidence from 2020–2025 (Fig 1A and 1B). However, whereas the status quo scenario was predicted to increase genotypic MQ resistance by 2025 in both Eastern and Western Cambodia, the triple ACT was predicted to reverse this trend, leading to declines in genotypic MQ and PPQ resistance (Fig 1C–1F). Genotypic joint resistance to both MQ and PPQ was not predicted to take off under either the status quo or the triple ACT scenario (Fig 1G and 1H). The similar trends in incidence despite differing resistance patterns likely reflect the fitness costs of resistance and the low assumed probability of phenotypic drug resistance given genotypic resistance markers.

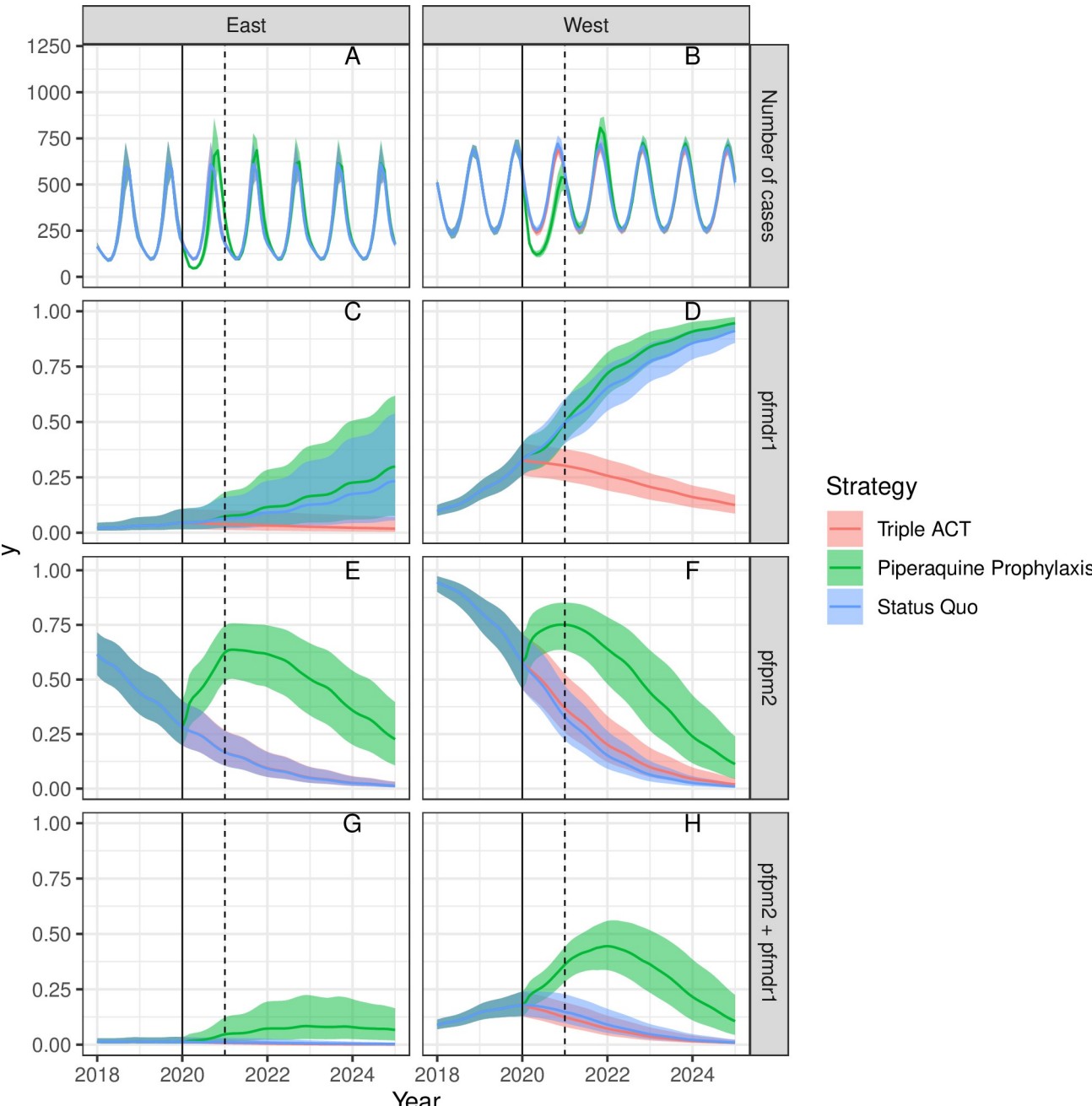

**Fig 1. Baseline results of model, 2018–2025.** Interventions begin in 2020 (solid line). The average duration of chemoprophylaxis is one year (dotted line at 2021). Results are shown for number of cases per month (Fig 1A and 1B), the proportion of new infections with multiple copy number *pfmdr1*, i.e. genotypic MQ resistance (Fig 1C and 1D), the proportion of new infections with multiple copy number *pfpm2*, i.e. genotypic PPQ resistance (Fig 1E and 1F), and the proportion of new infections with multiple copy numbers of both *pfpm2* and *pfmdr1*, i.e. genotypic resistance to both MQ and PPQ (Fig 1G and 1H).

Compared to the triple ACT scenario, the DP chemoprophylaxis scenario was predicted to have an immediate effect on malaria incidence (Fig 1A and 1B), assuming an initial coverage of 50%. However, these beneficial impacts do not extend beyond the average duration of prophylaxis of one year, and in fact a rebound in malaria incidence can be seen afterwards above

that predicted under the status quo scenario. The prophylaxis scenario is also predicted to increase the prevalence of genotypic PPQ resistance during its implementation phase (Fig 1E and 1F); although PPQ resistance declines again following cessation of prophylaxis, it remains at levels higher than that of the status quo or triple ACT scenario through 2025. The chemoprophylaxis scenario is predicted to lead to slight increases in genotypic MQ resistance (Fig 1C and 1D) and joint resistance to MQ and PPQ (Fig 1G and 1H). In sensitivity analyses, we found that even extended durations of chemoprophylaxis (10 years) could lose impact within 1–2 years due to the increases in PPQ resistance (S1 Appendix).

## Sensitivity analyses: Triple ACT scenario

We conducted multiple sensitivity analyses to better understand why resistance to PPQ and MQ failed to take off under the triple ACT scenario, despite increases in MQ resistance under the status quo scenario and both PPQ and MQ resistance under the prophylaxis scenario.

First, we increased the fitness of strains genotypically resistant to both MQ and PPQ to be just below the minimum fitness of single resistance to MQ and PPQ (i.e. the maximum possible value that would not produce inaccurate spread of joint resistance prior to 2020). Second, we decreased the probability of treatment success for joint resistance treated with the triple ACT to equal the maximum probability of treatment success given single resistance and single drug treatment. We thus assessed the most extreme values possible within the restrictions that jointly resistant strains not be more fit than singly resistant strains, and the triple ACT not be less successful at treating jointly resistant strains than a single drug is at treating single resistance. The results of these two changes, applied simultaneously, are shown in Fig 2. Following these changes, the status quo scenario still led to increases in MQ resistance and declines in PPQ resistance; however, substantial declines in PPQ resistance did not occur until MQ resistance was already widespread. Under the triple ACT scenario, MQ resistance and PPQ resistance both declined, though the rates of predicted decline were slow in Western Cambodia, with levels of joint MQ/PPQ resistance remaining roughly stable. Note that the estimated levels of MQ and PPQ resistance in Western Cambodia here are both higher than the baseline scenario, and represent a worse fit to the data (S1 Appendix Section 2).

We hypothesized that the reason joint MQ/PPQ resistance did not take off was related to our assumption that having multiple copy number *pfmdr1* and *pfpm2* led to phenotypic PPQ and MQ resistance, respectively, with probabilities that differed from one another but were fixed throughout the simulation period. Furthermore, these values were held fixed prior to fitting the fitness cost for each strain. To test this hypothesis, we fixed the probability of treatment success under each genotype-drug pair to be equal to a single value (0.625) intermediate to those used in the initial simulations and then re-fit the model. We then assessed the results of this refitted model under the baseline assumption as well as with the addition of the two sensitivity analyses above.

With this intermediate value of treatment success and the baseline parameters (Fig 3), joint resistance again remained roughly stable under the triple ACT scenario, with gradual declines in both PPQ and MQ resistance. Substantial increases in MQ resistance and faster declines in PPQ resistance were predicted under the status quo scenario. However, triple ACT treatment led to significant decreases in malaria transmission in Western Cambodia, likely reflecting the greater effects of *pfmdr1* on phenotypic MQ resistance and the greater expected levels of genotypic MQ resistance in 2020 prior to initiation of triple ACT treatment.

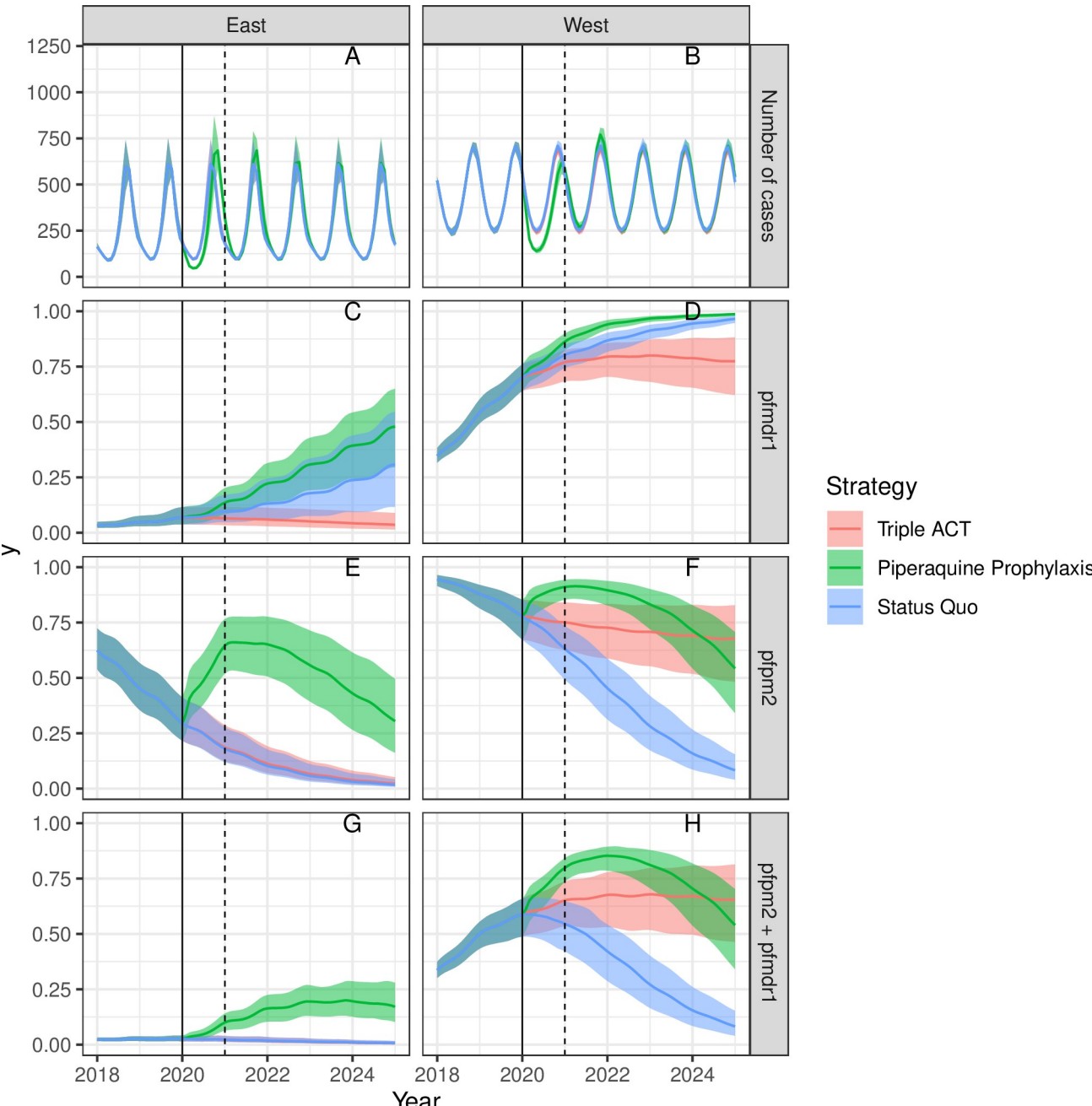

**Fig 2. Sensitivity Analysis 1—increased fitness and decreased probability of treatment success given joint resistance, 2018–2025.** Interventions begin in 2020 (solid line). The average duration of chemoprophylaxis is one year (dotted line at 2021).

Fig 4 shows a version of the model that includes all factors favoring the spread of joint MQ/PPQ resistance under the triple ACT scenario described above. Compared to the baseline scenario, we forced *pfmdr1* and *pfpm2* to have equal probabilities of producing phenotypic resistance; increased the fitness of jointly resistant strains; and decreased the probability of treatment success given joint genotypic resistance and triple ACT treatment. Indeed under this scenario triple ACT treatment increases the levels of joint MQ/PPQ resistance compared to both the status quo scenario and 2020 values.

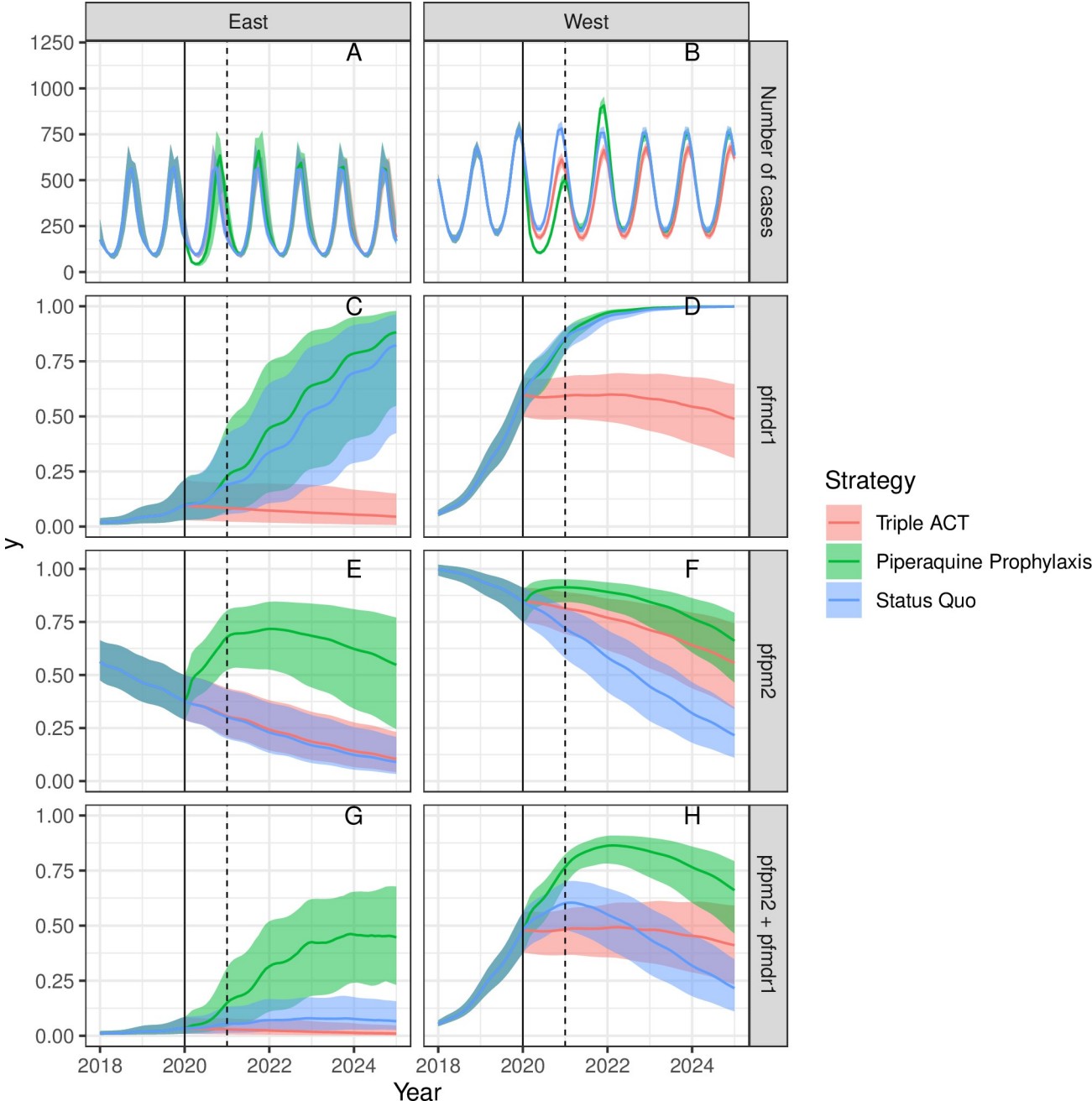

**Fig 3. Sensitivity Analysis 2 –equal probability of DP and ASMQ treatment success with *pfpm2* and *pfmdr1*, respectively, 2018–2025.** Interventions begin in 2020 (solid line). The average duration of chemoprophylaxis is one year (dotted line at 2021).

## Discussion

We have presented a model of how use of a triple ACT or chemoprophylaxis could affect malaria incidence and resistance to MQ and PPQ in Cambodia in the absence of other interventions. Under the initial assumptions and parameters, use of a triple ACT had minimal impact on overall malaria incidence but reversed the spread of mefloquine resistance predicted under the status quo scenario. In sensitivity analyses, it was possible but difficult to create a

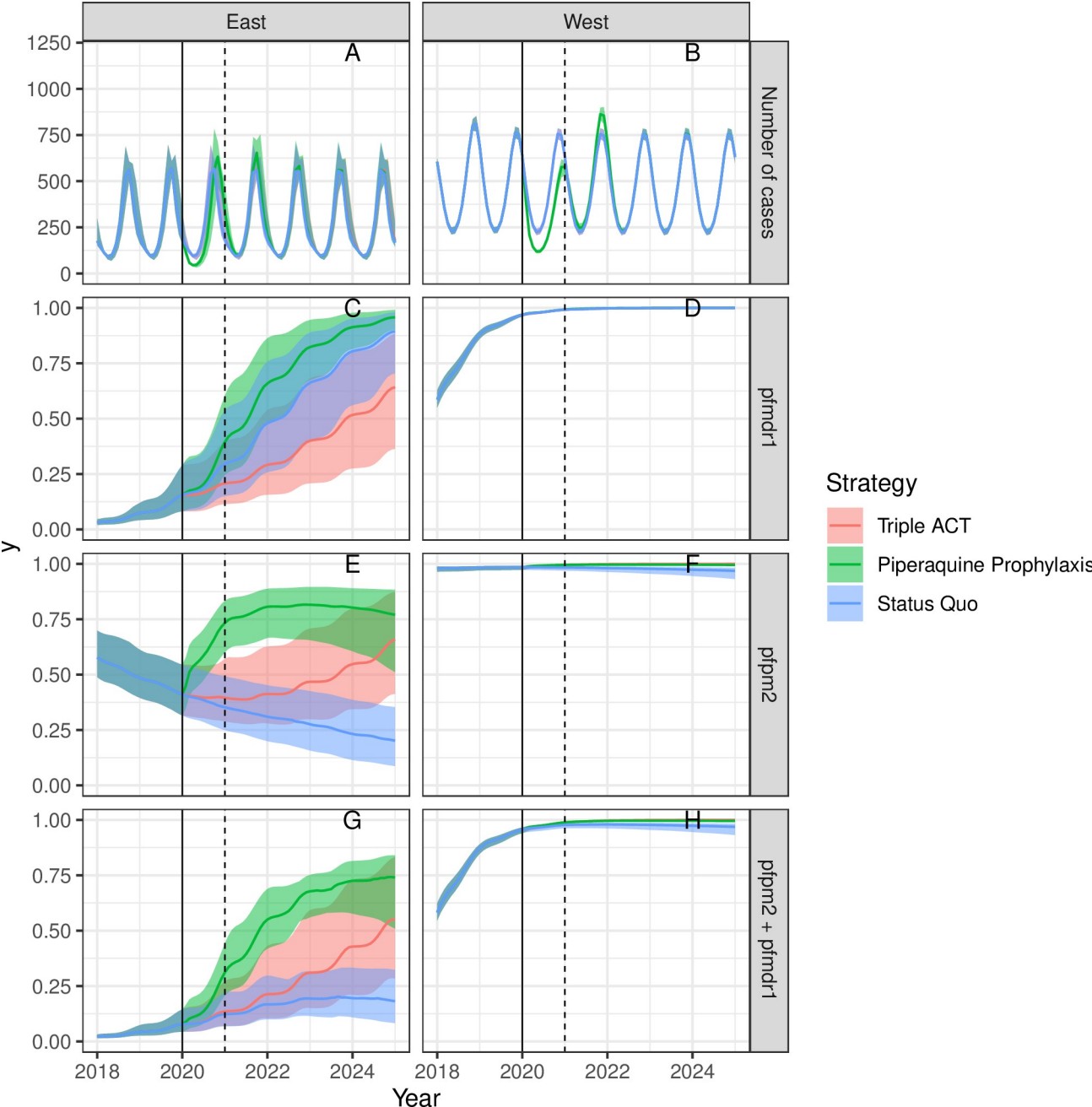

**Fig 4. Sensitivity Analysis 3—the fitness of jointly resistant strains is increased; the probability of DP-MQ successfully treating jointly resistant strains is decreased; and the probability of successful treatment of resistant strains with DP is forced to equal that with ASMQ.** Interventions begin in 2020 (solid line). The average duration of chemoprophylaxis is one year (dotted line at 2021).

situation under which the triple ACT scenario increased joint MQ/PPQ resistance substantially beyond its current levels. In contrast, we predicted that chemoprophylaxis could lead to significant declines in malaria incidence, but likely increase joint MQ/PPQ resistance as well; both of these effects of chemoprophylaxis were expected to reverse after cessation of chemoprophylaxis.

Why did the triple ACT lead to stable or declining levels of PPQ and MQ resistance, including joint resistance, under almost all parameters explored? Previous researchers have hypothesized that *pfpm2* and *pfmdr1* may have antagonistic effects [19]; notably, this was not encoded directly in this model. However, as joint resistance has seen only limited emergence in Cambodia until now, we did not allow the fitness of joint resistance to exceed that of either single resistance; furthermore, we assumed that triple ACT treatment would be at least as effective against genotypic joint resistance as both ASMQ and DP against strains with genotypic resistance to MQ and PPQ, respectively. In other words, joint resistance fails to take off as it is the "worst of both worlds", with low fitness and high probability of treatment success even under the triple ACT scenario.

A recent clinical trial found DP-MQ to be highly efficacious and safe in Cambodia, including in areas with high levels of resistance to DP [9]. Combined with its favorable impact on resistance in this model, triple ACT treatment could be an appealing choice in this area. However, caution should still be exercised. Our model assumes that the fitness of resistant strains is fixed over time, which may not be the case and could underestimate future resistance [20]. Of particular concern is the spread of malaria with *pfcrt* conferring PPQ resistance not included in this model. The GMS has proven past models and theories wrong, including those that initially predicted resistance to artemisinin would be avoided by use of ACTs [21]. Use of counterfeit, substandard, or inappropriate drugs may have contributed to past emergence of resistance in this region [22], and drug use and quality should be monitored closely. Treatment efficacy studies and genotypic resistance surveillance are also of utmost importance.

Chemoprophylaxis of high-risk populations including forest goers is also being explored in Cambodia. Although researchers have recognized DP as an appealing drug for chemoprophylaxis [11], current studies are focusing on other drugs including ASMQ and artemether-lumefantrine for reasons including preexisting resistance to DP. Introducing new drugs (such as artemether-lumefantrine or artesuante-pyronaridine) into the GMS as chemoprophylaxis could increase the initial efficacy of chemoprophylaxis. However, these alternative ACTs also have major drawbacks. Lumefantrine has a relatively short half-life and more complicated dosing schedule compared to piperaquine or mefloquine; it is also the primary partner drug used in Africa, which would heighten concerns about resistance. The safety of repeated courses of pyronaridine is currently unknown, and it represents a possible drug of last resort without pre-existing resistance in the GMS [11]. Additionally, as this model shows, chemoprophylaxis could also carry significant risks of drug resistance; unless such efforts succeed at rapidly eliminating malaria, they could risk further limiting the number of effective ACTs available in this region. This model also suggested there could be a rebound in malaria incidence post-chemoprophylaxis; similar increases in malaria incidence to or even above pre-intervention levels have been observed following mass drug administration in Cambodia and elsewhere [23,24].

Safety is another consideration when combining or repeating antimalarial use, as in the scenarios here. A clinical trial of DP-MQ found that the rate of clinical adverse events was not significantly increased compared to DP. Furthermore, the observed increase in QT interval with DP-MQ was not greater than that with DP [9]. Safety concerns are amplified when an antimalarial is given as chemoprophylaxis, due to its repeated and more widespread use in apparently healthy individuals. Existing data support the safety of monthly DP as intermittent preventive treatment or chemoprophylaxis [25]. Cost-effectiveness is another aspect that could be considered, as well as the infrastructure needed to implement each intervention (ex. could existing village malaria workers dispense monthly chemoprophylaxis, or is another system needed?)

There are some differences between the behavior predicted by our model and that observed in Cambodia for the period of 2018–2020. Limited published data were available to inform the model trends in genotypic MQ and PPQ resistance after the first-line treatment in Cambodia

switched to DP around 2017, but the data available thus far suggest the qualitative trends predicted here (i.e. declines in multiple copies of pfpm2 and increases in multiple copies of pfmdr1) are occurring in Cambodia as predicted, although the rates of spread may differ. Unlike in our model, Cambodia has seen rapid declines in P falciparum malaria cases since 2018. This timing has corresponded to an increase in other interventions such as use of mobile malaria workers and crackdowns on illegal logging, which are not reflected in this model. As such, this model should not be regarded as making quantitative predictions for malaria incidence in Cambodia.

The primary limitations of this model relate to its simplicity. We did not include resistance to artemisinin in our model, as such resistance is already widespread in Cambodia. We were primarily interested in understanding trends in partner drug resistance, as this can lead to treatment failure and thus affects the choice of first-line treatment. Additionally, we focused only on *pfmdr1* and *pfpm2*, whereas other genes such as *pfcrt* are increasingly understood to play a role in resistance to partner drugs. Besides resistance, the model provides an oversimplified view of immunity, the role of asymptomatic and submicroscopic infections, and population mixing patterns that may affect its results. However, the main qualitative findings of this model were consistent across two (separately fit) regions of Cambodia and multiple sensitivity analyses. Furthermore, the simplicity of the model allowed us to more easily isolate the role of individual parameters in understanding model output.

Overall, we conclude that triple ACTs may be useful at limiting spread of resistance to artemisinin partner drugs in high risk areas like the GMS. However, they could also slow declines in pre-existing resistance compared to single use of another drug. Furthermore, a switch to triple ACT treatment alone would not itself be sufficient for malaria elimination from this region. Chemoprophylaxis could accelerate malaria elimination but its effects are temporary and pose a higher risk of resistance. As a result, declines in malaria incidence in Cambodia since 2018 likely reflect the impact of new interventions not included in this model (ex. improved treatment coverage by village malaria workers and mobile malaria workers, crackdowns on illegal logging). A combined intervention strategy is likely the best option for achieving rapid malaria elimination from the GMS.

## Methods

### Modeling malaria transmission

To simplify the model, within each region (East and West) we assumed all cases occur within a single high-risk population. Conceptually, we could consider this group to consist of forest goers and residents of forested villages with active transmission. Only this high-risk population was modeled explicitly, though larger population denominators were used as needed to compare results of the model to data sources.

The model structure was informed by previous malaria models including [23,26,27] and was intentionally kept simple to facilitate qualitative understanding of resistance dynamics. Mosquitoes were modeled explicitly and could be in one of three disease states: susceptible to malaria, exposed to malaria and not yet infectious, and infectious with malaria. With regards to malaria infection and immunity, humans could belong to the following mutually exclusive states: S, susceptible to malaria (non-immune); E, exposed to malaria from this non-immune state, but not yet infectious; $I_s$, symptomatic, infectious malaria; $R_t$, post-treatment for malaria (no longer infectious, and protected from re-infection by the treatment drug); R, recovered and partially immune to malaria; and $I_a$, asymptomatic, infectious malaria (Fig 5).

We made the simplifying assumption that all non-immune individuals who develop malaria are symptomatic, and all partially immune individuals are asymptomatic. Individuals who are

A

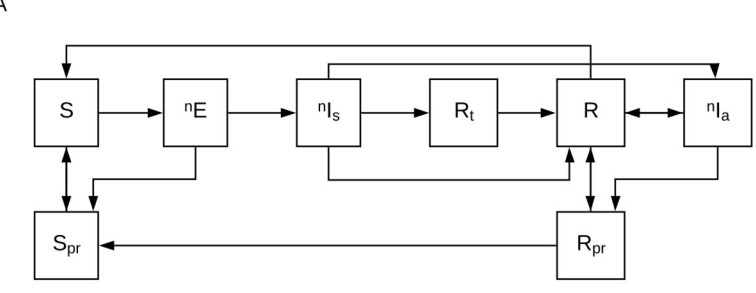

B

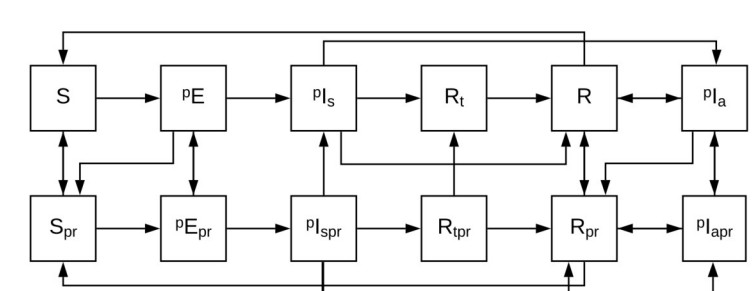

**Fig 5. Model states and transitions related to drug susceptible and genotypically PPQ-resistant infections.**
Mosquitoes are not shown. The disease states are as follows: S, susceptible to malaria; E: exposed to malaria, not yet infectious: $I_s$: infectious and symptomatic, $R_t$: recovered from malaria (i.e. no longer infectious) and protected by reinfection by prophylactic effect of treatment drug; R: recovered from malaria with partial immunity; $I_a$: infectious and asymptomatic. Subscript pr denotes prophylaxis. Superscript n denotes no drug resistance; superscript p denotes genotypic PPQ resistance. Entry to and exit from the model population, acquisition of resistance, and superinfection/recombination not shown.

symptomatic could receive either correct first-line treatment or other treatment (e.g. artemisinin monotherapies or other non-recommended treatments, with proportions varying over time as described in the S1 Appendix parameter tables). We allowed transmission from both symptomatic and asymptomatic states, but with differing probabilities (see S1 Appendix parameter tables). We assumed that partial immunity is temporary.

## Modeling partner drug resistance

With respect to drug resistance, we did not explicitly model resistance to artemisinin. Two forms of genotypic drug resistance were explicitly modeled: *pfmdr1* copy number ≥2 (conferring phenotypic MQ resistance with some probability) and *pfpm2* copy number ≥2 (conferring phenotypic PPQ resistance with some probability) [28,29]. Genotypic drug resistance was assumed to be transmissible between individuals, but phenotypic drug resistance was not, such that the probability that genotypic drug resistance would confer phenotypic drug resistance did not change over time. Phenotypic drug resistance was defined as late treatment failure to either ASMQ or DP. Those with late treatment failure re-enter the $I_S$ compartment immediately; the delay in recrudescence was not explicitly modeled. Individuals with a previous late treatment failure were assumed to have the same probability of future treatment success as all other individuals with the same genotype (treatment failure history was not tracked). Resistance was initialized in the population at a low level, and individuals receiving treatment without pre-existing genotypic drug resistance were allowed a low probability to spontaneously acquire such resistance. Genotypic drug resistance was assumed to confer a fitness cost, which

was included in the model as a reduced probability of transmission from mosquitoes to humans. These parameters were assigned wide prior ranges and inferred based on the data.

The model thus included four resistance strains, based on genotypic resistance to MQ, PPQ, or both. We tracked only a single dominant infection for each individual. Superinfection of asymptomatically infected individuals and recombination were included in the model (see S1 Appendix). Fig 5 shows a subset of the model states: those pertaining to drug susceptible infections (Fig 5A) and those pertaining to genotypically PPQ-resistant infections (Fig 5B).

## Modeling interventions

The triple ACT and chemoprophylaxis scenarios are options that have not yet been implemented in Cambodia, or are being implemented only in small trials. Therefore, it was necessary to consider a range of sensitivity analyses when considering these interventions. Both scenarios were modeled as being implemented from the beginning of 2020.

The triple ACT scenario was modeled similar to other treatment scenarios, with no changes in access. The main difference was that the triple ACT was assumed to be fully effective for all parasites having either a single copy number of *pfpm2* or a single copy of *pfmdr1*. For those parasites with multiple copy numbers of both *pfpm2* and *pfmdr1*, the treatment was also assumed to be effective with some probability. At baseline, we assumed that phenotypic resistance to PPQ and MQ are independent, such that P(resistance to DP-MQ|multiple copy numbers of both *pfpm2* and *pfmdr1*) = P(resistance to ASMQ|multiple copies of *pfmdr1*)*P (resistance to DP|multiple copies of *pfpm2*). This assumption was modified in sensitivity analyses.

Under the chemoprophylaxis scenario, a fixed proportion (50%) of the high-risk population received DP as chemoprophylaxis. We assumed the average duration of chemoprophylaxis was one year (i.e. repeated administration of DP throughout one year). We assumed enrollment occurred over a period of one month.

We assumed that symptomatic malaria infection was ruled out prior to administering chemoprophylaxis (as the first-line treatment in Cambodia is currently ASMQ, not DP), and that prophylaxis was not administered to those already treated within the last month. Chemoprophylaxis has the following effects on malaria parasites without genotypic PPQ resistance in the model. First, such parasites cannot infect susceptible or recovered humans receiving chemoprophylaxis. Second, individuals receiving chemoprophylaxis when already exposed to such parasites return to susceptible without becoming sick or infectious or developing immunity. Third, individuals receiving chemoprophylaxis when asymptomatically infected with such parasites clear their infection and return to the recovered state. Regarding those parasites with genotypic resistance to PPQ, chemoprophylaxis with DP is assumed to have the same probability of effectiveness as treatment. When chemoprophylaxis is effective, the effects of chemoprophylaxis are the same as those listed above. When it is not, infections and progression of infections occur as if no chemoprophylaxis were present.

## Parameterization and initialization

Parameters were chosen based on a review of the literature. For Eastern and Western Cambodia, seven of the most uncertain parameters were inferred from trends in malaria cases and resistance over time. Prior distributions were set for these parameters and posterior distributions were derived via Incremental Mixture Importance Sampling [30]. This procedure was done separately for Eastern and Western regions. The S1 Appendix contains figures comparing the model results under the posterior distributions to the data. The results reported in the

main text are based on 500 random draws from the posterior distribution of parameters (main results), and 200 random draws for the sensitivity analyses.

Changes in malaria trends over time were captured in three ways in the model. First, we assumed that the proportion of symptomatic malaria cases receiving appropriate treatment increased over time. Second, we assumed that the malaria transmission parameter declined over time (considering, e.g., increase in coverage of LLHINs). Third, we assumed that the population at risk of malaria declined over time (considering deforestation and urbanization in Cambodia). More detail on the parameters involved in these assumptions is available in the S1 Appendix.

Initial conditions were derived by running the model with an initial seed of 100 infected humans and 1% infected mosquitoes for 5 years to reach near-equilibrium conditions. This initialization was performed with no drug resistance. In Western Cambodia, mefloquine resistance was input into the model beginning in 2000 with 15% of new infections having genotypic MQ resistance. In Eastern Cambodia, due to insufficient data and apparent low levels, this was maintained as 0 until 2010, at which point there was assumed to be 10% genotypic MQ resistance. In both regions, 1% genotypic PPQ resistance was introduced at the time of the switch from ASMQ to DP.

## Supporting information

**S1 Appendix. Supplemental methods and results.**
(DOCX)

## Acknowledgments

We would like to thank Benoit Witkowski for helpful discussions regarding the development and current status of drug resistance in Cambodia.

## Author Contributions

**Conceptualization:** Amber Kunkel, Michael White, Patrice Piola.

**Data curation:** Amber Kunkel.

**Formal analysis:** Amber Kunkel.

**Investigation:** Amber Kunkel.

**Methodology:** Amber Kunkel, Michael White.

**Supervision:** Michael White, Patrice Piola.

**Visualization:** Amber Kunkel.

**Writing – original draft:** Amber Kunkel.

**Writing – review & editing:** Amber Kunkel, Michael White, Patrice Piola.

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
