## [Decision Letter · Decision Letter 0]

5 Oct 2020

Dear Dr Kunkel,

Thank you very much for submitting your manuscript "Novel anti-malarial drug strategies to prevent artemisinin partner drug resistance: a model-based analysis" for consideration at PLOS Computational Biology.

As with all papers reviewed by the journal, your manuscript was reviewed by members of the editorial board and by several independent reviewers. In light of the reviews (below this email), we would like to invite the resubmission of a significantly-revised version that takes into account the reviewers' comments.

In addition to addressing the reviewers' comments, please provide the code in a more suitable manner, such as through a GitHub repository.

We cannot make any decision about publication until we have seen the revised manuscript and your response to the reviewers' comments. Your revised manuscript is also likely to be sent to reviewers for further evaluation.

Sincerely,

Alex Perkins

Associate Editor

PLOS Computational Biology

Nina Fefferman

Deputy Editor

PLOS Computational Biology

Reviewer's Responses to Questions

**Comments to the Authors:**

Reviewer #1: PCOMPBIOL-D-20-01377: Novel anti-malarial drug strategies to prevent artemisinin partner drug resistance: a model-based analysis

The emergence of drug resistance is a real threat to malaria transmission and studies that try to understand its spread are of paramount importance. This work attempts to model the potential impacts of anti-malarial drug strategies to understand artemisinin partner drug resistance using data from Cambodia. The main text is well written and ease follow as well as the model schematic shown in Figure 1. However, by reading the Supporting Information Section we find many essential information about methods and data, and notice that the model presented is much more complex than what is presented in the main text. The authors performed too much data manipulation, stated too many assumptions, and designed a model with too many parameters (obtained from other articles or assumed values for 24 fixed parameters, plus 9 time-dependent parameters and fitted the model for another 8 parameters) that is difficult to not be sceptical about inferences made. Moreover, the reported monthly cases model output from Western Cambodia presents a poor fitting. The same poor fitting is observed for the prevalence (PCR) plots.

Reviewer #2: The authors have conducted a no trivial work aiming to contain the spread of artemisinin’s partners drugs resistance. Their final goal seems to prevent the spread of artemisinin’s resistance outside the GMS to Africa by preventing its partner drug resistance.

The authors use a simple compartmental seir model for malaria transmission to assess the capacity of 3 scenarios (current “ASMQ treatment”, triple-ACT “DP+MQ” and treatment + prophylaxis “ASMQ+DP”) to contain the partner drug resistance.

The modeling framework seems appropriate with fair assumptions. The paper is well written but should be improved.

Major comments

1) Baseline results

Please add 2018-2020 incidence / resistance data if any to figure 2.

2) According to the authors, Cambodia has several times switched between ASMQ and DP and currently back

to ASMQ. The authors believe ASMQ efficacy could be short-lived triggering their work to support novel treatment strategies. Since ASMQ and DP have inconsistent, why not test/suggest new drugs? The authors alluded for instance to AL as a possible candidate and I wonder why ASMQ and DP remain their preferred drugs in their suggested treatment scenarios. I am just trying to get more background / literature regarding the choice of treatment strategies / drugs by the authors.

3) Cambodia is not a high burden country; the urgency remains elimination rather than changing policy to reduce incidence.

Why is prophylaxis important here? Maintaining ASMQ+DP in the paper does not add any value to the purpose of this work since rather than preventing partner drugs’ resistance, the prophylaxis is increasing the spread of such resistance. To me the prophylaxis's strategy simply adds more complexity and distraction to the paper with no benefit that all.

Since the main purpose of their work is to prevent resistance, the authors could for instance test and suggest a treatment/prophylaxis strategy that reduces both incidence and resistance, or simply leave it out if not effective, or please justify why it is so crucial to this paper.

4) I have a problem with the “Rate of waning protection by treatment drug”. The parameter

(w_t) value seems generic and attributed to all drugs. Artemisinin and partner drugs have different half-lives and for that reason, a clarification should be given to how PKPD model parameters were fitted to drugs' half-lives. If there are assumptions on half-lives, it should be clearly stated in the text but it can't be 20days for artemisinin and partner drugs.

Additionally, this (drug kinetics) may be useful to sensitivity analysis to assess differential spreads of resistance between treatment strategies but I leave it to the authors to decide whether they want to pursue and exploration.

5) The discussion section will benefit from a paragraph on safety / toxicity of triple-ACT or combination of treatment / prophylaxis, and possibly a sentence on cost-effectiveness.

Minor

1) Would be more informative to the reader to have some numbers / statistics about artemisinin

resistance and its partner drugs resistance in the GMS, added to the background/introduction.

2) Fig 1a is not cited in the text.

Reviewer #3: In this study the authors develop a model, parameterised to reflect malaria transmission in Cambodia, to explore the impact of triple artemisinin combination therapies (ACTs) and chemoprophylaxis in limiting spread of drug-resistant parasites and accelerate elimination. This is an interesting subject to explore in detail. The complexity of the interplay between drug-sensitive & drug resistant parasites has been simplified in the model, but I think it has been done in an appropriate way, given the aims of the study. The results obtained in the baseline model, that triple ACT therapy could reverse mefloquine resistance, but not impact overall incidence, and the chemoprophylaxis did impact incidence but encouraged the spread of drug-resistant parasites, are intuitive but important to explore in detail. Furthermore, the sensitivity analysis, which explores the changes in model assumptions required for triple ACT treatment to promote spread of parasites resistant to both mefloquine and piperaquine is very useful. I have some minor comments that I would like to be addressed, and I am happy to recommend the work for publication once this has been done.

1) Lines 103-105. The authors state that partner drug resistance more important in driving treatment failure than artemisinin resistance. Would they be able to provide a reference for this?

2) Line 146: Define DS. Drug susceptible?

3) Regarding treatment coverage: parameter ‘app’ governs the proportion of symptomatic individuals receiving first line treatment. What proportion of symptomatic individuals receive non-recommended treatment (e.g. monotherapy)? If it is (1-app), is it realistic to assume that 100% of symptomatic infections are treated? If not, what would the implications be for the modelling results presented here?

4) Regarding Figure 1, the upper schematic implies that state nIa (asymptomatic, infectious individuals) can only be reached through the recovered state, R. Presumably this is incorrect?

5) Figure 2 legend: I’m not sure called treatment with a triple ACT “Joint Treatment” is sensible, but I leave this choice to the authors.

6) The authors demonstrate that the chemoprophylaxis scenario decreased malaria incidence, but the policy is only explored for 1 year. It would be interesting to see what happens if the policy is pursued e.g. do these decreases continue, or does the increased prevalence of drug-resistant malaria eventually render the intervention ineffective?

**Have all data underlying the figures and results presented in the manuscript been provided?**

Reviewer #1: None

Reviewer #2: Yes

Reviewer #3: Yes

PLOS authors have the option to publish the peer review history of their article (what does this mean?). If published, this will include your full peer review and any attached files.

Reviewer #1: No

Reviewer #2: **Yes: **ANDRE LIN OUEDRAOGO

Reviewer #3: No
---

## [Decision Letter · Decision Letter 1]

31 Jan 2021

Dear Dr Kunkel,

Thank you very much for submitting your manuscript "Novel anti-malarial drug strategies to prevent artemisinin partner drug resistance: a model-based analysis" for consideration at PLOS Computational Biology. As with all papers reviewed by the journal, your manuscript was reviewed by members of the editorial board and by several independent reviewers. The reviewers appreciated the attention to an important topic. Based on the reviews, we are likely to accept this manuscript for publication, providing that you modify the manuscript according to the review recommendations.

Please add some text to the manuscript to convey the spirit of your response to the initial review by Reviewer 1. Addressing the new comments by Reviewer 1 would also be appreciated. We hope to evaluate this next round of revisions editorially (and relatively quickly) if these remaining issues can be addressed satisfactorily.

Sincerely,

Alex Perkins

Associate Editor

PLOS Computational Biology

Nina Fefferman

Deputy Editor

PLOS Computational Biology

[LINK]

Reviewer's Responses to Questions

**Comments to the Authors:**

Reviewer #1: PCOMPBIOL-D-20-01377R1: Novel anti-malarial drug strategies to prevent artemisinin partner drug resistance: a model-based analysis

The work attempts to model the potential impacts of anti-malarial drug strategies to understand artemisinin partner drug resistance using data from Cambodia. Surprisingly, the overall malaria incidence trend presented in Figures 2-5 does not change over time and across different scenarios. If we look at Figure 2, for example, under the Status Quo scenario, the model output indicates that the number of cases per month in West Cambodia does not increase when the proportion of new infections with multiple copy number pfmdr1 goes from around 0.1 in 2018 to around 0.9 in 2025. Figures 3 and 4 bring similar results. If we take Figure 5, even when the proportion of new infections with multiple copy number pfmdr1, pfpm2 or pfmdr1+pfpm2, is close to 1 (panels D, F and H), the overall malaria incidence predicted by the model also does not increase over the time. As mentioned on the text, if all innervations are held constant and those with late treatment failure re-enter the I_s compartment immediately, why the spread of artemisinin partner drug resistance does not affect the overall malaria?

As we can find in Figure 1, individuals move to compartment R after recovering and become partially immune to malaria. Based on a strong assumption, the model assumes that all partially immune individuals (from compartment R) who develop malaria are asymptomatic (lines 361-362) and move to compartment I_a. After recovering from the asymptomatic infection, individuals move back to compartment R (recovered and partially immune to malaria). The rate of waning immunity (parameter “w”) is w=1/365 day^(-1), that is, after recovering from an infection (symptomatic or asymptomatic), it takes on average 365 days before any individual becoming “fully” susceptible again (compartment S). Considering that the model assumes a single high-risk population, I wonder if the rate to move from R to I_a is not considerably higher than the rate to move from R to S (waning immunity). In affirmative case, it is expected that, under model assumptions, most individuals will be “trapped” between compartments R and I_a. As a consequence, the overall malaria incidence (I am assuming only symptomatic cases) might be sensible to variations on the rate of waning immunity. In order to clarify this point, it would be interesting to have a plot of proportion of individuals in compartments S, I_s, I_a and R over time. I also wonder if the authors do not consider to perform a sensitivity analysis of the rate of waning immunity (parameter “w”). Please, clarify this point.

Overall, the authors concluded that triple ACTs may be useful at limiting spread of resistance to artemisinin partner drugs (lines 321-322). Although pfpm2 and pfpm2+pfmdr1 results from Figure 2 are pretty much the same for triple ACT and Status Quo regimes, the first performs better for pfmdr1. However, if my understanding is correct, sensitivity analysis results raise questions about the model robustness with respect to the conclusion and do not clearly favour triple ACT regime. We can note in Figure 3 for West Cambodia that triple ACT regime performs worse than the Status Quo regime for pfpm2 and pfpm2+pfmdr1. For a long-term analysis, the same behaviour can be found in Figure 4. Further, the authors say: “In sensitivity analyses, it was possible but difficult to create a situation under which the triple ACT scenario increased joint MQ/PPQ resistance substantially beyond its current levels” (lines 239-241). Regarding joint MQ/PPQ resistance, it seems to me that results from Figures 2-5 suggests that, overall, the Status Quo scenario performs better than the triple ACT. Please, clarify this point.

Posterior distribution bounds of parameters “beta_min_m”, “p” and “f_p” are out of prior distribution bounds (Supporting Information, line 87). Please, clarify this point.

Reviewer #2: The revision by the authors significantly improved the paper. Most comments have been appropriately addressed.

No further comments.

Reviewer #3: I am satisfied that the authors have dealt with my comments appropriately. I'm happy to recommend publication

**Have all data underlying the figures and results presented in the manuscript been provided?**

Reviewer #1: None

Reviewer #2: Yes

Reviewer #3: None

PLOS authors have the option to publish the peer review history of their article (what does this mean?). If published, this will include your full peer review and any attached files.

Reviewer #1: No

Reviewer #2: **Yes: **Andre Lin Ouedraogo

Reviewer #3: No
---

## [Editor Report · Decision Letter 2]

3 Mar 2021

Dear Dr Kunkel,

We are pleased to inform you that your manuscript 'Novel anti-malarial drug strategies to prevent artemisinin partner drug resistance: a model-based analysis' has been provisionally accepted for publication in PLOS Computational Biology.

Best regards,

Alex Perkins

Associate Editor

PLOS Computational Biology

Nina Fefferman

Deputy Editor

PLOS Computational Biology

---

## [Editor Report · Acceptance letter]

13 Mar 2021

PCOMPBIOL-D-20-01377R2 

Novel anti-malarial drug strategies to prevent artemisinin partner drug resistance: a model-based analysis

Dear Dr Kunkel,

I am pleased to inform you that your manuscript has been formally accepted for publication in PLOS Computational Biology. Your manuscript is now with our production department and you will be notified of the publication date in due course.

With kind regards,

Alice Ellingham
